# Quality of Life and Working Conditions of Hand Surgeons—A National Survey

**DOI:** 10.3390/medicina59081450

**Published:** 2023-08-11

**Authors:** Léna G. Dietrich, Esther Vögelin, Michael J. Deml, Torsten Pastor, Boyko Gueorguiev, Tatjana Pastor

**Affiliations:** 1Department for Plastic and Hand Surgery, Inselspital University Hospital Bern, University of Bern, 3012 Bern, Switzerland; lena.dietrich@insel.ch (L.G.D.); esther.voegelin@insel.ch (E.V.); tatjana.pastor@insel.ch (T.P.); 2Department of Sociology, Institute of Sociological Research, University of Geneva, 1211 Geneva, Switzerland; michael.deml@unige.ch; 3Division of Social and Behavioural Sciences, School of Public Health, University of Cape Town, Cape Town 7925, South Africa; 4Department of Orthopaedic and Trauma Surgery, Lucerne Cantonal Hospital, 6002 Lucerne, Switzerland; torsten.pastor@luks.ch; 5AO Research Institute Davos, 7270 Davos, Switzerland

**Keywords:** hand surgeons, quality of life, Swiss, Switzerland

## Abstract

*Background and Objectives*: Providing high-quality care for patients in hand surgery is an everyday endeavor. However, the quality of life (QoL) and working conditions of hand surgeons ensuring these high-quality services need to be investigated. The aim of this study was to evaluate the QoL and working conditions of Swiss hand surgeons. *Materials and Methods*: A national survey with Swiss hand surgeons was conducted. Standardized questionnaires were completed anonymously online. Core topics included working conditions, QoL, satisfaction with the profession, and aspects of private life. *Results*: A total of 250 hand surgeons were invited to participate, of which 110 (44.0%) completed the questionnaire. Among all participants, 43.6% stated that they are on call 4–7 days per month, versus 8.2% never being on call. Overall, 84.0% of the residents, 50.0% of the senior physicians, 27.6% of the physicians in leading positions, and 40.6% of the senior consultants/practice owners, as well as 55.1% of the female and 44.3% of the male respondents, felt stressed by their job, even during holidays and leisure time. Out of all participants, 85.4% felt that work affects private relationships negatively. Despite the reported stress, 89.1% would choose hand surgery as a profession again. Less on-call duty (29.1%) and better pay (26.4%) are the most prioritized factors for attractiveness of a position at a hospital. *Conclusions*: The QoL of Swiss hand surgeons is negatively affected by their workload and working hours. Residents, senior physicians and female surgeons suffer significantly more often from depression, burnout or chronic fatigue in comparison to leading positions, senior consultants/practice owners and male surgeons. Better pay or less on-call duty would make the work more attractive in acute care hospitals.

## 1. Introduction

Since January 2015, hand surgery has been recognized as an independent specialist title by the Federal Council in Switzerland. In addition to a rapid development in surgical subjects, the increase in the population and of hand injuries and injuries to the peripheral nerves were certainly decisive factors causing changes in the profession as a hand surgeon. But what about the satisfaction and quality of life of Swiss hand surgeons now, 7 years later?

In today’s world, “work–life balance”, career opportunities, satisfaction and equality are all core issues for the survey of quality of life as a surgeon. Cathelain et al. used an assessment to survey the quality of life of gynaecologists in France [1]. The study showed clear gender differences in the impact of workload on the time they have for themselves, their families and friends. The job can have a significant psychological impact, and burnout rates are high [1]. In an American study, “work–life integration” was examined as an important contribution to quality of life [2]. It also showed that gender and ethics have a complex influence on burnout rates [2].

The work–life balance and well-being of surgeons have recently emerged as new and important areas of research [2]. This is one important way to maintain and ensure a high standard of patient care. Physicians’ burnout rates and the rate of sleep disorders of surgeons are higher than in the general population [3]. Patient safety is meticulously studied and implemented. But what about the health risk of those persons treating patients? Deficits are already evident in their basic human needs; regular meals and physical exercise are rare [3].

Given these considerations, we aimed to evaluate the quality of life and working conditions of Swiss hand surgeons. Therefore, an online survey for hand surgeons in Switzerland was developed and conducted, as this approach has been already established in the literature [4]. The results of the survey could inform about the development of interventions to improve quality of life and working conditions for hand surgeons in Switzerland.

## 2. Materials and Methods

### 2.1. Survey Development

An online survey to assess quality of life and working conditions among hand surgeons in Switzerland was developed. Furthermore, potentially relevant topics and constructed questions were assembled. In discussions within our multidisciplinary team covering hand surgery and sociology, we condensed the survey with the goal of making it concise and appropriate for hand surgeons. LimeSurvey (LimeSurvey GmbH, 2003, GPL, JavaScript, PHP) was used as online platform. The survey was translated from English into both German and French by bilingual research team members. Subsequently, surveys were administered in these three official languages in Switzerland. Respondents were able to choose their preference at the beginning of the survey. We piloted the survey in all three languages with participants from a target group, and discussed and adjusted its wording accordingly in each separate language. In the current study, we presented all study results in English.

### 2.2. Survey Content

The survey consisted of 38 questions and the core topics were working conditions, quality of life, satisfaction with the profession and aspects of private life. The collected data set was analysed, implementing univariable and multivariable models. The survey contained questions addressing different aspects of quality of life, including satisfaction as a Swiss hand surgeon in relation to: (1) profession/career; (2) family life; (3) social life; (4) free time; and (5) general satisfaction. Background questions (gender, age, status relationship, children, position and additional designation) were included.

Differences between the different specifications (gender, age, position, place of work, chronic fatigue/burnout/depression) were evaluated. Furthermore, the relationships between the occurrence of chronic fatigue/burnout/depression and (1) gender, (2) position and (3) aspects of quality of life were analysed.

Standardised instruments established in the literature were used to measure issues such as satisfaction [1,4]. A 5-point Likert scale was used to assess participants’ satisfaction regarding wages, family life, social life, job/career (1: “not satisfied at all” to 5: “completely satisfied”) and quality of life (1: “insufficient“ to 5: “excellent”).

Participants were asked (1) whether they would study medicine again, and (2) whether they would choose hand surgery as a profession again. We also asked about personal experiences considering burnout, chronic fatigue, or depression. Respondents chose from a list of factors that would potentially make a job at an acute care hospital more attractive and created their own ranking.

### 2.3. Survey Administration

The survey was available online from 1 March to 15 May 2022 and was completed by most of the participants within 8–12 min. All residents and senior physicians from the mailing lists of the Swiss Society for Hand Surgery and the Association of Young Hand Surgeons were invited. Membership of these societies is not compulsory in Switzerland, but we assumed that most Swiss hand surgeons are members of at least one of these two organizations. The societies invited their members to participate via email. We asked the societies to send a reminder email to their members 1 or 2 weeks after the first invitation to increase the response rate. Participation in the survey was voluntary and anonymity was always maintained.

### 2.4. Statistical Analysis

Statistical analysis was performed using IBM SPSS software package (Version 28, IBM, SPSS, Armonk, NY, USA). All observations were reported as percentage rates. Spearman’s correlation and chi-square tests were applied to screen for associations between each pair of categorical variables.

## 3. Results

### 3.1. Respondents

A total of 110 participants responded to the survey, including 25 (22.7%) residents and 85 (77.3%) other attendees. The response rate was 44.0% (110 respondents among 250 society members). The demographic characteristics of the participants are presented in Table 1. The respondents replied in all three official languages in Switzerland: German (n = 85, 77.3%), French (n = 23, 20.9%) and English (n = 2, 1.8%). Most participants were in the 30–44 years age category (n = 60, 54.6%). There were more male (n = 61, 55.5%) than female (n = 49, 44.5%) participants. The men were more often in leading positions, working as senior consultants or higher (n = 42, 68.9%), in contrast to female hand surgeons, where 19 (38.8%) were residents. Most respondents were in a relationship (n = 101, 91.8%) and 77 (70.0%) had children.

### 3.2. Professional Details/Working Hours

In total, 34 (30.9%) participants from a category A clinic (level I), 39 (35.5%) participants from a category B clinic (level II), 7 (6.4%) participants from a category C clinic (level III), 26 (23.6%) participants from a practice and 4 (3.6%) participants from another clinic (non-teaching institution) participated in the study. In terms of positions, 25 (22.7%) residents, 24 (21.8%) senior physicians, 29 (26.4%) physicians in leading positions, and 32 (29.1%) senior consultants/practice owners responded. The work-related characteristics of the participants are presented in Table 2. Most participants were on call at least 4–7 days per month (n = 48, 43.6%), only 9 (8.2%) were never on call. Average actual working time per week differed from working time per week according to the contract and ranged between 30 and 82 h. Holidays ranged from 8 to 60 days per year. Working hours during free time ranged from 0 to 30 h. There were more physicians in leading positions and senior consultants/practice owners compared to residents and senior physicians. Among all responders, 65 (59.1%) aspired to work in a managerial position, 60 (54.6%) stated that more satisfaction could be achieved through more autonomous work, and 98 (89.1%) replied that they would choose hand surgery as a profession again.

### 3.3. Health Status

A total of 40 (36.4%) of the respondents reported 7 h of sleep per night on average, while 32 (29.1%) replied that they had already suffered or have been suffering from burnout, chronic fatigue or depression. A total of 10 participants (9.1%) were smokers, and 3 (2.7%) reported use of microstimulates or amphetamines. During holidays and/or free time, almost half of the respondents (n = 54, 49.1%) felt stressed or affected by work. In detail, 21 (84.0%) of the residents, 12 (50.0%) of the senior physicians, 8 (27.6%) of the physicians in leading positions, and 13 (40.6%) of the senior consultants/practice owners, as well as 27 (55.1%) of the female and 27 (44.3%) of the male respondents, felt stressed by the job during holidays and leisure time. A total of 99 (90.0%) of the respondents stated that they were understood and supported by their environment (e.g., family, friends, workplace) when it came to professional matters. In total, 12 (48.0%) of the residents, 8 (33.3%) of the senior physicians, 4 (13.8%) of the physicians in leading positions and 8 (25.0%) of the senior consultants/practice owners suffered from chronic exhaustion, burnout or depression. The analysis related to gender revealed that 22 (44.9%) of the female and 10 (16.4%) of the male hand surgeons suffered from depression, burnout or chronic exhaustion. Women and residents were therefore more likely to suffer from depression, chronic fatigue or burnout (gender: *p* < 0.001, position: *p* = 0.018).

### 3.4. Quality of Life and Satisfaction

Respondents generally rated their quality of life as insufficient in 1 (0.9%) case, moderate in 12 (10.9%) cases, good in 22 (20.0%) cases, very good in 66 (60.0%) cases and excellent in 9 (8.2%) cases (Table 3). The most important aspect of life for 70 (63.6%) respondents was family, followed by job/career (n = 21, 19.1%), time for themselves (n = 13, 11.8%), and social life (n = 6, 5.4%).

The results for the different aspects of satisfaction in relation to social life, family life, job and career, and salary are summarized in Table 4. In total, 43 (39.1%) of the respondents rated their satisfaction in terms of social life as moderate, 20 (18.2%)—as insufficient and 2 (1.8%)—as having no social life at all (not satisfied at all). A total of 49 (44.5%) of the participants rated their satisfaction in terms of family as good, with 29 (26.4%), 9 (8.2%) and 4 (3.6%) rating it as moderate, insufficient or not satisfied at all, respectively. Most respondents rated their satisfaction in terms of job as good (n = 69, 62.7%), 15 (13.7%)—as moderate, 4 (3.6%)—as insufficient, and 1 (0.9%)—as not sufficient at all. A total of 30 (27.3%) of the respondents rated their satisfaction in terms of salary as moderate, 29 (26.4%)—as insufficient, and 5 (4.5%) were not satisfied at all. Out of all participants, 85.4% felt that work affects their private relationships negatively.

Eighty-two (74.5%) respondents stated that they would study medicine again. The specialisation of hand surgery would be pursued again by 98 (89.1%) respondents. In total, 5 (20.0%) of the residents, 5 (20.8%) of the senior physicians, and 3 (9.4%) of the senior consultants/practice owners claimed a moderate or poor quality of life, while 24 (82.8%) of the physicians in leading positions were satisfied. Forty-three (39.1%) respondents stated that they had already seriously thought about giving up the profession.

### 3.5. Significant Correlations

There was a significant correlation between depression, chronic fatigue, and burnout with impaired quality of life (*p* = 0.028). Those respondents who declared that they suffered from work-related stress during holidays and leisure time were significantly more likely to suffer from depression, chronic fatigue and burnout (*p* < 0.001). The factor work-related stress during holidays and leisure time correlated significantly with age (*p* = 0.004) and position (*p* < 0.001).

Among the participants, 21 (84.0%) of the residents, 12 (50.0%) of the senior physicians, 8 (27.6%) of the physicians in leading positions and 13 (40.6%) of the senior consultants/practice owners declared that they suffer from work-related stress during their holidays and leisure time, indicating a significant relationship between the participants’ position and their work-related stress during holidays and leisure time; the respondents in lower positions were significantly more stressed than their colleagues in higher positions, *p* < 0.001.

### 3.6. Factors That Would Make Employment at an Acute Care Hospital for Hand Surgery More Attractive

In the design of the survey, we provided a list of options from which participants could select a preferred factor that would make their job in an acute care hospital more attractive. All available factors, namely better compensation, fewer services, part-time, regulation of time recording and compensation, more autonomy, more supervision and mor part-time research/part-time clinic, were included and used in the participants’ rankings. The preferred factors are depicted in Figure 1. A total of 32 (29.1%) participants would prefer to have less on-call services and 29 (26.4%) would prefer to receive higher payment. When only the two most important factors for increase of job attractivity in an acute care hospital were considered, the most important ones included better payment for 108 (98.2%) respondents and having less on-call time for 62 (56.4%) respondents.

## 4. Discussion

The current study evaluated a survey about quality of life and working conditions of hand surgeons in Switzerland. The main findings were as follows. (1) A large majority of Swiss hand surgeons are stressed and suffer under long working hours; despite a proportion of participants who do not suffer from depression, chronic fatigue or burnout, these participants expressed high workloads which impairs the quality of life. (2) Female hand surgeons suffer significantly more frequently from depression or burnout symptoms, that might be a problem in the near future due to the increasing number of women working in medicine. (3) Young surgeons are at risk of being stressed during their holidays and leisure time; however, the Swiss Society of Surgery of the Hand depends on a motivated and healthy young generation.

### 4.1. Quality of Life

Aspects of quality of life (work, family, social life) differ depending on age and position of hand surgeons (residents, senior physicians, physicians in leading position, senior consultants/practice owners). The current study revealed that especially younger female surgeons (residents, younger senior physicians) suffer significantly more due to stressful working conditions followed by a reduced quality of life. However, the quality of life of surgeons is an extremely important and decisive factor. This can be explained by the statement of Balch et al.: “As surgeons, we need to set an example of good health to our patients and future generations of surgeons”. As Balch et al. already published: “To provide the best care for our patients, we need to be alert, interested in our work, and ready to provide for our patient’s need” [5]. Individuals, societies and organisations should implement strategies and programs that promote work–life integration [2].

### 4.2. Burnout

The world health organization (WHO) defined burnout as follows: “Burn-out is a syndrome conceptualized as resulting from chronic workplace stress that has not been successfully managed. It is characterized by three dimensions: (1) feelings of energy depletion or exhaustion; (2) increased mental distance from one’s job, or feelings of negativism or cynicism related to one’s job; and (3) reduced professional efficacy”. A meta-analysis including 16 cross-sectional studies and a total of 3581 subjects concluded that about 3% of surgeons suffer from an extreme form of burnout (burnout syndrome) and up to 34% of surgeons may experience burnout characterised by high levels of burnout in one of the three subscales (emotional exhaustion, depersonalization, and personal accomplishment) [6]. A study in the USA from 2010 examined the correlation between burnout and medical errors and reported that major medical errors by surgeons are strongly related to a surgeon’s degree of burnout and mental quality of life [7]. In the current survey, the numbers and burnout rates among hand surgeons (29.1%) are similar to those reported by Pulcrano et al. [8] and higher in comparison to the general Swiss population (18%) [9]. A lack of control over one’s work life can contribute as a crucial factor to burnout [10]. Residents and younger senior physicians are at an increased risk for burnout, depression and chronic fatigue and are more likely to report a poor quality of life in comparison to surgeons in higher positions (leading position, senior consultant/practice owner) [11]. Data demonstrate that burnout is more prevalent in early career stages [11]. Residents and fellows are more frequently affected by burnout than senior doctors and age-matched graduates [12]. Fainberg et al. discuss low-cost measures, such as structured mentoring programmes for residents, which could both prevent burnout and promote departmental culture. More control over working hours can improve work–life balance. Furthermore, trainees affected by depression or suicidality must have unfettered access to mental health resources. There is a need to reduce the stigma associated with asking for help [12]. Chesak et al. identified specific interventions to prevent burnout in female physicians: (1) removing barriers to career satisfaction, work–life integration and mental health; (2) identifying and reducing gender and maternal biases; (3) mentoring and sponsorship opportunities; and (4) family leave, breastfeeding and childcare policies and support [13].

There is a correlation between quality of patient care and the prevalence of burnout among surgeons [14]. In a meta-analysis, Hocking et al. report that burnout among physicians is associated with poor functioning and sustainability of health care organisations. Burnout leads to the professional exit and a high turnover of physicians, and has been shown to affect the quality of patient care. Health organisations should invest more time and effort in implementing evidence-based strategies to curb physician burnout, especially among physicians in residency training [15]. It is also well known that gender has a complex impact on work–life integration [2]. Akazawa et al. argue for support systems, such as education systems, mentorships and promotions, to enable female doctors in academic positions [16]. A mentorship program for young female residents has been recently introduced by the University of St. Gallen in cooperation with multiple university hospitals in Switzerland, fostering leading positions at hospitals or scientific careers for young women.

### 4.3. Influence of COVID-19 Pandemic

A recently published study reports that women experienced more stress than men during the COVID-19 pandemic (regardless of parental status) and suggests that more than the added demands of childcare contributed to the gendered differences in the experience of stress during the pandemic [17]. This is in contrast to a 2020 study from the USA which showed decreased burnout and high career satisfaction rates among neurosurgeons during the COVID-19 pandemic [18]. Assuming that elective operations were cancelled due to the COVID-19 pandemic, it can be concluded that a reduced workload was present during this period. Indirectly, the correlation between a lower workload and a decrease in burnout rates can be observed.

### 4.4. Gender

It is well known that gender has a complex impact on work–life integration [2]. Regarding the recruitment and success of women in academic surgery, women have some differences (access to collaborations and support, issues in balancing family and work life, and the degree to which perceptions are changing). It is important to further understand women’s perceptions of their role in academic surgery and to address obstacles that exist for both men and women [19]. Women report significantly lower career satisfaction in comparison to men (77% versus 82%) [20]. However, when it is considered that female surgeons were less likely to be satisfied with their career, as published in other reports [21], the current study shows an equal trend of satisfaction. Baptise et al. showed that women are more likely to be married to a professional who works full-time than men (90% versus 37% for specialists; 82% versus 41% for residents) and are less likely to have a permanent job. Women are more likely to be responsible for childcare planning, meal planning, grocery shopping and holidays planning. Gender-neutral tasks included financial planning and paying monthly bills. Female surgeons are more likely to live with a full-time spouse and are primarily responsible for household management. The aim should therefore be to even out these gendered differences [22].

### 4.5. Working Hours

Our national survey included questions regarding working hours. Previous studies reported that time spent at the workplace has an influence on medical mistakes and depression and burnout rates [3,23]. Tibble et al. demonstrated that surgeons are more than twice as likely to attract complaints as their physician peers. Factors that could have influenced these results are involvement in surgical procedures and treatments [24]. This might be explained by the prevalence of stressful, overbooked days and less time for patient care outside the operating room. These factors are also reflected in the results of the current study and 42% of the participants would like to spend fewer hours at work.

### 4.6. Career Satisfaction

Johnson et al. stated that work–life integration conflicts contribute to career dissatisfaction. Moreover, collegial support of work–life integration efforts had the strongest association with higher career satisfaction [20]. This suggests an important need for additional efforts to strengthen collegial support in medical teams. The current study showed a promising trend, as 99 (90.0%) of the respondents stated that they experience support from their environment, although this was not defined more precisely (colleagues, family, friends). This fact might play a crucial role in terms of career satisfaction. Among the respondents of the current study, 12 (10.9%) would no longer choose hand surgery as a profession. This rate is similar to the reported numbers in the literature (21% female general surgeons and 13% male general surgeons) [21]. Cathelain et al. reported that three quarters of respondents would choose the same job again if they could, which was interpreted as evidence of career satisfaction [1].

### 4.7. Working Conditions

More than four-fifths of the residents (84%) reported that they were feeling stressed during leisure time and holidays in the current study. In contrast, 50% of the senior physicians, only 28% of the physicians in leading positions, and 40% of senior consultants/practice owners were stressed by their job during holidays and leisure time. Bohrer et al. reported that this requires a concerted action of all relevant parties (hospital administration, societies, insurance companies) to improve the working condition for surgeons [3]. In addition to legal adjustments, every surgeon might take measures to improve the quality of life in daily practice (preserving medical attitudes against the constraints of economics, taking responsibility to create a good working climate, and proper training [3]).

### 4.8. Attractivity of Acute Care Hospitals

We identified several potential variables, such as better compensation, fewer services, more part-time work, less regulation of time recording and compensation, more autonomy, more supervision and more part-time research/part-time clinic, to increase attractivity for hand surgeons in acute care hospitals. In general, all proposed aspects were of high interest. In particular, reducing on-call time and increasing payment should be taken into consideration in order to increase the attractivity for hand surgeons working in acute care hospitals, as most respondents in this cohort agree that these two factors would make an acute care hospital more attractive. However, due to financial pressure in the health system, this will be difficult to realize.

In summary, as already stated by Anderson et al., the focus should be on work–life integration instead of work–life balance [2]. Work–life integration is a desirable goal in different professions. A stable work–life integration would promote well-being, productivity, satisfaction and patient care [2]. The term work–life balance, as discussed in the introduction, entails inconsistency when talking about working conditions. Work should not be the counterpart to life and an integration of work into life seems to be desirable.

### 4.9. Strengths and Limitations

A strength of the current study is the large sample size of respondents across the country (N = 110), with a satisfactory response rate of 44.0% (compared to other similar studies) [21]. Novel insights into potential factors influencing the quality of life of hand surgeons in Switzerland were provided, which is a field of research that has previously received no attention in Switzerland. As with all survey methods, there is a potential for recall bias in surgeons’ responses. The Appendix A was not validated by statistical tests. Some questions left room for interpretation regarding the direction of quality of life (e.g., what social life means in detail). No more detailed descriptions of each aspect were provided in the current survey. Furthermore, the COVID-19 pandemic could have influenced the results of the current study. Future work should address these issues by specifying such factors and providing rigorous validation of the survey constructs.

Overall, as Switzerland has already one of the best health care systems worldwide, the implementation of the two most important factors that influence the quality of life and working conditions of Swiss hand surgeons—namely, less on-call duties and better payment—would be costly. Another solution could be the reduction in on-call time for hand surgeons. Hand surgery teams are usually small, and the on-call time is generally higher for hand surgeons compared to doctors working in bigger teams. Not all minor hand injuries must be treated by certified hand surgeons at night. A possible solution could also be the integration of general surgeons and emergency doctors into the treatment of minor hand injuries at night. This could significantly reduce the time that certified hand surgeons who are on call must spend on site at the hospital without generating additional costs on technical upgrades or salary. However, this measure must be implemented without a decrease in treatment quality. Better training of general surgeons and emergency doctors could be a possible solution.

## 5. Conclusions

The quality of life of hand surgeons in Switzerland is negatively affected by their workload and working hours. Residents, senior physicians and female surgeons suffer significantly more often from depression, burnout or chronic fatigue in comparison to leading positions, senior consultants/practice owners and male surgeons. Better pay or less on-call duty would make the work more attractive in acute care hospitals. Despite the captured difficulties, 89.1% of respondents would choose hand surgery as a profession again.

## Figures and Tables

**Figure 1 medicina-59-01450-f001:**
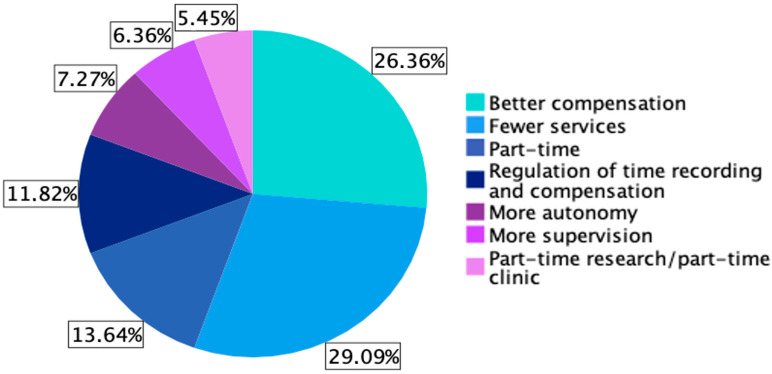
Reported preferred factor that would make a job in an acute care hospital more attractive.

**Table 1 medicina-59-01450-t001:** Demographic participants’ characteristics.

Characteristic	Number (Rate)
**Age Group** (years)	
18–29	4 (3.6%)
30–44	60 (54.6%)
45–60	34 (30.9%)
60+	12 (10.9%)
**Gender**	
Male	61 (55.5%)
Female	49 (44.5%)
**Language**	
German	85 (77.3%)
French	23 (20.9%)
English	2 (1.8%)
**Relationship**	
Yes	101 (91.8%)
No	9 (8.2%)
**Children**	
Yes	77 (70.0%)
No	33 (30.0%)

**Table 2 medicina-59-01450-t002:** Work-related participants’ characteristics.

Characteristic	Number (Rate)
**Position**	
Resident	25 (22.7%)
Senior physician	24 (21.8%)
Leading position	29 (26.4%)
Senior consultant/practice owner	32 (29.1%)
**Hospital/Practice Category**	
Level I	34 (30.9%)
Level II	39 (35.5%)
Level III	7 (6.4%)
Practice	26 (23.6%)
Other	4 (3.6%)

**Table 3 medicina-59-01450-t003:** Quality of life.

Quality of Life	Number (Rate)
Insufficient	1 (0.9%)
Moderate	12 (10.9%)
Good	22 (20.0%)
Very good	66 (60.0%)
Excellent	9 (8.2%)

**Table 4 medicina-59-01450-t004:** Participants’ satisfaction.

Category	Number (Rate)
**Social Life**	
Not satisfied at all	2 (1.8%)
Insufficient	20 (18.2%)
Moderate	43 (39.1%)
Good	35 (31.8%)
Completely satisfied	10 (9.1%)
**Family Life**	
Not satisfied at all	9 (8.2%)
Insufficient	29 (26.4%)
Moderate	49 (44.5%)
Good	19 (17.3%)
Completely satisfied	4 (3.6%)
**Job/Carreer**	
Not satisfied at all	4 (3.6%)
Insufficient	15 (13.7%)
Moderate	69 (62.7%)
Good	21 (19.1%)
Completely satisfied	1 (0.9%)
**Salary**	
Not satisfied at all	5 (4.5%)
Insufficient	29 (26.4%)
Moderate	30 (27.3%)
Good	29 (26.4%)
Completely satisfied	17 (15.4%)

## Data Availability

The datasets used and/or analyzed during the current study are available from the corresponding author on reasonable request.

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
