# Peer review of "Quality of Life and Working Conditions of Hand Surgeons—A National Survey"

_medicina, 2023, doi:10.3390/medicina59081450_

Round 1

Reviewer 1 Report

interesting article, it helps the reader to compare his experiences to the community of hand surgeons to understand if their condition reflects community condition.

At the time it was proposed it may have been influenced, in 2022, by the covid 19 pandemic.

About the survey content

How many questions have the survey?

Line 57 please consider avoid repeat differences.. different

Lines 306-330

Please consider to show the WHO definition of burnout

As Authors reported the percentage of burnout in surgeon, please consider to report burnout date in general Swiss population

Figure 1 I see 29% of surgeons would like to have fewer services 13% would like part time. I understand that 42% would like to spend at work less hours, please consider to report in conclusions

Reviewer 2 Report

The study of Dietrich et al was conducted in an effort to assess the living and working conditions of Swiss hand surgeons, a relatively small group of specialized surgeons, known for their refined surgical techniques and commitment to high-quality treatments. The study was conducted in a standardized questionnaire nationwide (Switzerland), evaluating aspects of private life as well as the working environment, quality of life, and professional satisfaction.

Overall, the study is well conducted, and presents interesting findings for surgeons with increased burden of shifts, need to deliver high-quality results, with the main focus on "functionality". The study indicates clearly the issues related to the performance of this demanding profession and highlights the parameters requiring re-assessment, in order to improve the physician`s satisfaction and subsequently enable a sustainable delivery of healthcare services. 

The only concerns relate to the usage of a nonvalidated questionnaire, which could have been avoided. Nevertheless, the authors stated this aspect clearly in their limitations. The language used is appropriate and the statistical methods appear sound.

In my view, the study does not require any revisions. 
